# Experimental Study: The Effect of Pore Shape, Geometrical Heterogeneity, and Flow Rate on the Repetitive Two-Phase Fluid Transport in Microfluidic Porous Media

**DOI:** 10.3390/mi14071441

**Published:** 2023-07-18

**Authors:** Seunghee Kim, Jingtao Zhang, Sangjin Ryu

**Affiliations:** 1Department of Civil and Environmental Engineering, Nebraska Center for Materials and Nanoscience, University of Nebraska-Lincoln, Lincoln, NE 68588, USA; 2Department of Civil and Environmental Engineering, University of Nebraska-Lincoln, Lincoln, NE 68588, USA; jingtao.zhang@huskers.unl.edu; 3Department of Mechanical and Materials Engineering, Nebraska Center for Materials and Nanoscience, University of Nebraska-Lincoln, Lincoln, NE 68588, USA; sangjin.ryu@unl.edu

**Keywords:** porous-media compressed-air energy storage (PM-CAES), underground hydrogen storage (UHS), fluid flow in porous media, repetitive drainage–imbibition cycles, microfluidics technology, PDMS model

## Abstract

Geologic subsurface energy storage, such as porous-media compressed-air energy storage (PM-CAES) and underground hydrogen storage (UHS), involves the multi-phase fluid transport in structurally disordered or heterogeneous porous media (e.g., soils and rocks). Furthermore, such multi-phase fluid transport is likely to repeatedly occur due to successive fluid injections and extractions, thus, resulting in cyclic drainage–imbibition processes. To complement our preceding study, we conducted a follow-up study with microfluidic pore-network devices with a square solid shape (Type II) to further advance our understanding on the effect of the pore shape (aspect ratio, Type I: 5–6 > Type II: ~1), pore-space heterogeneity (coefficient of variation, COV = 0, 0.25, and 0.5), and flow rates (Q = 0.01 and 0.1 mL/min) on the repetitive two-phase fluid flow in general porous media. The influence of pore shape and pore-space heterogeneity were observed to be more prominent when the flow rate was low (e.g., Q = 0.01 mL/min in this study) on the examined outcomes, including the drainage and imbibition patterns, the similarity of those patterns between repeated steps, the sweep efficiency and residual saturation of the nonwetting fluid, and fluid pressure. On the other hand, a higher flow rate (e.g., Q = 0.1 mL/min in this study) appeared to outweigh those factors for the Type II structure, owing to the low aspect ratio (~1). It was also suggested that the flow morphology, sweep efficiency, residual saturation, and required pressure gradient may not severely fluctuate during the repeated drainage-–imbibition processes; instead, becoming stabilized after 4–5 cycles, regardless of the aspect ratio, COV, and Q. Implications of the study results for PM-CAES and UHS are discussed as a complementary analysis at the end of this manuscript.

## 1. Introduction

Energy storage refers to the process of capturing energy produced at one time and storing it for later use. Energy storage is important because it enables better balancing of the supply of and the demand for energy, which is essential for a reliable and efficient energy system. That is, energy storage allows the excess energy produced during periods of low demand to be stored for use during periods of high demand, thereby reducing the need for new-power-generation capacity and helping to stabilize the grid. Energy storage is also important for enabling the integration of renewable energy sources, such as wind and solar power, into the currently existing energy system. These sources of energy can be variable, with energy production affected by weather conditions and the time of day. Energy storage can help to smooth out fluctuations in energy production, providing a more stable and predictable source of energy for consumers.

There are various types of energy storage technologies, including batteries, pumped hydro storage, compressed-air energy storage, hydrogen storage, flywheels, and thermal storage, among others [1]. Each technology has its own unique characteristics and applications, and the choice of technology will depend on factors such as the type of energy being stored, the scale of the storage system, and the cost.

Geologic subsurface energy storage, such as porous-media compressed-air energy storage (PM-CAES) and underground hydrogen storage (UHS), involves the multi-phase fluid transport in structurally disordered or heterogeneous porous media (e.g., soils and rocks; an example is shown in Figure 1) [2]. Furthermore, such multi-phase fluid transport is likely to repeatedly occur due to successive fluid injections and extractions, thus resulting in cyclic drainage–imbibition processes. In this regard, a comprehensive understanding of the repetitive multi-phase fluid transport in heterogeneous porous media is imperative to improve the efficiency of those operations. Nonetheless, there have been very limited experimental studies that attempted to examine such repetitive drainage–imbibition processes at the fundamental pore-scale level. Readers are referred to our preceding publication, Zhang et al. [3], for relevant previous studies that used microfluidic pore-network devices.

Recently, we experimentally examined such a repetitive two-phase fluid flow between water and mineral oil using microfluidic pore-network devices [3,5]. Even though these preceding studies provided detailed observations on the fluid flow during the repetitive drainage and imbibition processes, there were limitations in that the pore structure was either homogeneous [5] or only with a circular solid shape (Type I), which yields a large aspect ratio (the ratio of the width of the pore space to the width of the pore throat [3]. Here, we conducted a follow-up study with microfluidic pore-network devices with a square solid shape (Type II) to further advance our understanding on the effect of the pore shape, as well as pore-space heterogeneity and flow rates, on the repetitive two-phase fluid flow in general porous media. In this regard, this study using Type II geometry can be regarded as a companion study to that of Zhang et al. [3] which mainly utilized Type I geometry. The experimental design and test procedure are briefly described first, followed by the results and discussion in this manuscript.

## 2. Analysis Factors

### 2.1. Dimensionless Numbers

We use the capillary number (*Ca*) and viscosity ratio (*M*) to analyze the experimental results [6]:(1)$$Ca=\frac{V\cdot \mu_{nw}}{\gamma cos\theta }\;\mathrm{and}\;M=\frac{\mu_{nw}}{\mu_{w}}$$
where *γ* is the interfacial surface tension coefficient [mN/m], *θ* is the contact angle of a solid-fluid–fluid system [deg], *V* is the velocity of an advancing nonwetting fluid [m/s], and $\mu_{nw}$ and $\mu_{w}$ are the viscosity of nonwetting and wetting fluids [Pa·s], respectively. This study used the *Ca* and *M* values from previous and current PM-CAES projects collected by Zhang et al. [3] as a benchmark to design the experiment, based on the widely acknowledged notion that the fluid flow pattern in porous media will be similar if *Ca* and *M* are of similar magnitudes in a given operation [6,7].

### 2.2. Correlation Coefficient

To quantitatively evaluate the similarity of fluid flow patterns during repetitive drainage and imbibition steps in the micromodels, we utilized the correlation coefficient (*R*_c_). Our analysis involved examining the distribution of the nonwetting fluid in binary images obtained by applying MATLAB’s (R2022a, MathWorks) Otsu’s method to convert the original experimental images. *R_c_* is defined the same way as in our preceding study [3] as follows:(2)$$R_{c\;(i)}=\frac{\sum_{j=1}^{N}\left(I_{j,i}-\bar{I_{j,i}}\right)\left(I_{j,i+1}-\bar{I_{j,i+1}}\right)}{\sqrt{\sum_{j=1}^{N}{\left(I_{j,i}-\bar{I_{j,i}}\right)}^{2}\sum_{j=1}^{N}{\left(I_{j,i+1}-\bar{I_{j,i+1}}\right)}^{2}}},$$
where *I* is the value of a pixel (0 or 1), *i* is the step of either drainage or imbibition, *j* is the index of a pixel, *N* is the total number of pixels, and $\bar{I_{j,i}}$ is the mean of $I_{j,i}$.

### 2.3. Minkowski Functionals

The Minkowski functionals (e.g., *M*_0_, *M*_1_, *M*_2_, …) can provide information on the distribution and connectivity of fluids in porous media [8,9]. *M*_0_ and *M*_1_ in a two-dimensional (2-D) space are related to the measurements of the covered area (*F*) and boundary length (*U*) of an object as follows [8]:(3)$$M_{0}\left(A\right)=\int_{A}^{}d^{2}\vec{r}=F(A),$$
(4)$$M_{1}\left(A\right)=\frac{1}{2\pi }\int_{\partial A}^{}d\vec{r}=\frac{1}{2\pi }U\left(A\right),$$
where ∂A is the regular boundary of a domain (either the wetting or nonwetting fluid in this study) of an object *A*, and $d\vec{r}$ is an element of the surface of the domain *A*. Readers are referred to Zhang et al. [3] for more detailed explanations. For the discussion of the experimental results, we normalized the boundary length (*U*) in Equation (4) by the covered area (*F*) in Equation (3) to analyze the flow morphology of the nonwetting fluid more objectively; as such,
(5)$$m_{0}=\frac{U(A)}{F(A)}=\frac{{2\pi M}_{1}(A)}{M_{0}(A)}$$

The Minkowski functionals were calculated using MATLAB’s image processing code. The procedure involved cropping the images of the pore-network domain obtained after each drainage and imbibition step (refer to Section 5.1 and Section 5.2 for an example of images). These cropped images were then converted into binary images using the graythresh function in MATLAB. This function determined a global image threshold that minimized the intraclass variance between black and white pixels. The Minkowski functionals *F* and *U* of the nonwetting fluid were subsequently computed based on the number of pixels representing the nonwetting fluid, utilizing the built-in functions in MATLAB.

## 3. Review—Salient Flow Dynamics in Type I vs. Type II

### 3.1. Pore Shape

Hydrocarbon reservoirs and saline aquifers primarily consist of sedimentary rocks such as sandstones and carbonates. Compared with sandstones, carbonates have a higher occurrence of natural fractures, which contribute to secondary porosity and create pathways for fluid flow [10]. The size and shape of pores play a crucial role in determining the fluid flow within porous rocks [11], and the typical pore size in such rocks is at the scale of micrometers [12]. Considering these factors, we devised two distinct pore-network structures, referred to as Type I and Type II, for our experimental studies. Type I was designed to replicate the intergranular pores commonly found in sandstones [5,13] and has a randomized distribution of circular solids within the pore network. In contrast, Type II aimed to simulate naturally fractured carbonate rocks, as observed in studies by Warren and Root [14] and Soleimani [15]. In our study, the average aspect ratio of the inner pore diameter (D_2_ in Figure 2) to throat width (D_1_) was greater than 5 for Type I, while for Type II, it was approximately 1 (B_2_/B_1_ in Figure 2).

### 3.2. Drainage

For Type I, the interface between the wetting and nonwetting fluids exhibits dynamic evolution due to the prevalent occurrence of a Haines jump [5], which is common for a pore network with a large aspect ratio (i.e., pore size far greater than the throat width) [12,16]. Furthermore, the invasion of the nonwetting fluid tends to develop in both orthogonal directions more uniformly compared with that in Type II. In contrast, the piston type [17] is favored in the direction of the pressure gradient for Type II geometry. The interface between the wetting and nonwetting fluids maintains a stable evolution through the flow channels [12]. The leak mechanism, a phenomenon in which the nonwetting fluid in the adjacent pore space (or node) hinders the displacement of the wetting fluid trapped in a preceding channel, is also a feature for Type II. The trapped wetting fluid can leak out later depending on the drainage rate and fluid viscosity. If the drainage is too slow, the trapped wetting fluid may remain in place.

### 3.3. Imbibition

A phenomenon called “collapse in a channel” is frequently observed during the imbibition of the wetting fluid for Type I, along with the displacement of the nonwetting fluid in both orthogonal directions [5]. The “collapse in a channel” describes the phenomenon in which the wetting fluid flows along and around the wall of solid parts when the aspect ratio is large [16]; depending on pressure, the wetting fluid that accumulates on a wall of the channel may become unstable and fill the channel, disconnecting the nonwetting fluid (i.e., collapse) and leaving behind a portion of it in the preceding pore space (or node). This mechanism is also known as “pinch-off” [18].

On the other hand, three different phenomena, “Imbibition I1”, “Imbibition I2”, and “snap-off” [16,17] are more common during the imbibition process for Type II. “Imbibition I1” denotes that the nonwetting fluid is displaced by the wetting fluid throughout all possible flow paths in the pore space. The interface between the wetting and nonwetting fluids in the channel becomes unstable when the interface does not touch the wall of channels (i.e., only staying in the pore space), which results in a rapid displacement of the nonwetting fluid. “Imbibition I2” denotes when the nonwetting fluid meets a sharp edge at the intersection of the flow channels and then is split, resulting in shorter fluid bodies that facilitate the displacement of nonwetting fluid further away. The “snap-off” is the phenomenon in which the nonwetting fluid percolates through a channel while some wetting fluid still remains on the surface of the wall [17,19,20]. When the capillary pressure decreases and the frontal meniscus is not in the channel, the wetting fluid on the walls can swell to push the nonwetting fluid until the nonwetting fluid gets separated, leaving behind the separated nonwetting fluid bodies in the pore space.

## 4. Experimental Study

### 4.1. Pore Structure of the Micromodel

We developed structurally heterogeneous pore-network models of Type II based on the geometry and dimensions of our previously studied homogeneous micromodels [5]. In these models, the square solids had a width of 320 μm, while the pore throats had a width of 80 μm. The dimensions of the pore-network domains were 20 × 20 × 0.1 mm^3^, which were larger than the representative elementary volume of sedimentary rocks. The main domain of the pore-network consisted of 50 × 48 square solids spanning in a direction parallel and perpendicular to the pressure gradient, resulting in the generation of 51 × 47 pore throats (Figure 3).

The position of each solid was determined using a MATLAB code, where the width of pore throats followed a log-normal distribution with a mean value of *w* = 80 μm in both directions. The coefficient of variation (COV = σ_w_/*w*) for the pore-throat width was set to either 0, 0.25, or 0.5. To account for the resolution limit of the fabrication method, the minimum width of the pore throat was set at 20 μm. It was recommended that the height-to-width ratio of the solid part be maintained between 0.2 and 2 to avoid any potential defects during the PDMS-based microfluidic model fabrication [21,22]. With that, the height of the flow channel was chosen to be 100 μm. A summary of the geometric characteristics of the micromodels can be found in Table 1.

After obtaining the solid coordinates from the code, the information was utilized to replicate the pore structure using AutoCAD 2020 (Autodesk). For this study, two distinct pore structures were designed independently, corresponding to each COV value of 0, 0.25, and 0.5 (as shown in Figure 3). The fabricated micromodel devices had porosity values of 0.36 and 0.38 for COV = 0.25 and 0.5, respectively. Their permeability values were also determined through single-phase fluid flow tests (4.0–6.4 Darcy (=10^−12^ m^2^)) and were found to be in the same order.

### 4.2. Fabrication of the Pore-Network Micromodel

We replicated the microfluidics fabrication technique described by Zhang et al. [3] to create hydrophobic or oleophilic pore-network micromodels using polydimethylsiloxane (PDMS). Initially, we utilized MATLAB and AutoCAD to design the pore structure of the micromodels, incorporating an inlet, the main pore-network domain, and an outlet (refer to Figure 4; please note that the inlet and outlet were positioned based on the desired flow direction). To ensure a more uniform flow and pressure gradient within the main pore-network domain, we designed the inlet and outlet in a tree-like configuration. A transparent photomask with a resolution of 254,000 dots per inch (dpi) was then created according to the design, which served as the basis for producing a master mold through photolithography. This master mold consisted of a positive relief pattern of the SU-8 100 photoresist (MicroChem, Westborough, MA, USA) on a silicon wafer. Subsequently, we prepared PDMS (Dow Corning Sylgard 184) using the standard mixing ratio of 10:1, poured it onto the mold, degassed it, and cured it in an oven at 65 °C for 6 h. The resulting cured PDMS was carefully peeled off from the mold, and holes for the inlet and outlet were punched. It was then bonded to a separate PDMS layer formed on a glass slide using plasma activation. The fabricated device was post-cured for one day in the oven at 65 °C. Finally, we inserted a 2.5 cm-long tube (Tygon tubing, 0.02″ inner diameter × 0.06″ outer diameter) into the inlet and outlet holes.

### 4.3. Materials

The wetting and nonwetting fluids used in the experiments were mineral oil (Fisher Chemical BP2629-1) and deionized water, respectively, as indicated in Table 2. The water was colored using green food dye. We had measured the advancing contact angle at the water–oil interface via MATLAB-based image processing in our preceding study [5]. It was approximately 45° for both Types I and II geometries during the entire drainage–imbibition cycles. All experiments in this study were conducted at room temperature, which was approximately 25 °C. The calculated *Ca* values for this study were approximately 2.3 × 10^−6^ (log *Ca* ≈ −5.6) and 2.3 × 10^−5^ (log *Ca* ≈ −4.6) for the applied flow rates of 0.01 and 0.1 mL/min, respectively. The *M* value in this study was determined to be 2.38 × 10^−2^ (log *M* = −1.62). Both the log *Ca* and log *M* values closely align with those obtained from the expected PM-CAES operations summarized by Zhang et al. [3]. These *Ca* and *M* values suggest that the drainage process would either fall within the transition zone between capillary and viscous fingering (at 0.01 mL/min) or within the viscous fingering regime (at 0.1 mL/min) according to Zheng et al. [23].

### 4.4. Experimental Setup and Procedure

Before each test, the newly fabricated micromodel was saturated with mineral oil. We utilized a syringe (BD, 3 mL) and a high-precision syringe pump (PHD ULTRA) to maintain a constant flow rate (*Q*) of either 0.01 or 0.1 mL/min throughout ten repetitive drainage and imbibition cycles. The effluent fluid from these cycles was collected in an empty container placed at the outlet side. To ensure the percolation of approximately 2.5 pore volumes of the nonwetting or wetting fluid through the tested micromodel, the injection duration for both the drainage and imbibition processes was set at 330 s for *Q* = 0.01 mL/min and 33 s for *Q* = 0.1 mL/min.

To observe the fluid transport within the main pore-network domain during the experiment, a microscope (LEICA S90) and a high-speed camera (LEICA MIC 190 HD) were utilized. Two pressure transducers (OMEGA PX309-100GV, Bienne, Switzerland) with a pressure range of 0–689 kPa and a sensitivity of 19.96 Pa/kPa were connected to the right and left sides of the micromodel, respectively, to measure the fluid pressure throughout the test. The data acquisition process was automated using a data logger (Keysight 34972A, Rosebery, Austrilia), a computer, and a DC power supply (Keysight E3630A). The complete experimental setup is depicted in Figure 4.

To commence the repetitive flow test, the first step involved initiating drainage from the right-hand side of the micromodel. Drainage was sustained for the desired flow duration, starting from the moment the front of the nonwetting fluid passed through the entrance of the tree-like inlet. Subsequently, the syringe pump was halted, and both valves were promptly closed, indicating the completion of the first drainage step (right to left).

Subsequently, new outer tubes were introduced, replacing the existing ones. The left-side tube was filled with mineral oil, while the right-side tube was filled with water. The syringe pump, along with a new syringe filled with oil, was moved to the left side, and a fresh empty container was positioned on the right-hand side. The two valves were reopened, and the syringe pump was started, initiating the reoccupation of the micromodel’s pore space with mineral oil through a process known as forced imbibition. This imbibition phase continued for the same duration as the previous drainage step, starting from the moment the oil front passed through the entrance of the tree-like inlet. Once again, the two valves were closed using the same procedure, marking the completion of the first imbibition step (left to right). This entire sequence was repeated until a total of ten cycles were completed.

We performed repetitive drainage–imbibition tests for two iterations using a homogeneous model (COV = 0), fabricating a new model each time. Additionally, we conducted tests for four heterogeneous models, with two models having COV = 0.25 and the other two models having COV = 0.5. The Reynolds number for the fluid flow in the tree-like inlet was calculated to be within the range of 0.5–5 for drainage and 0.01–0.1 for imbibition. As a result, all flow experiments in this study fell within the laminar flow regime.

## 5. Experimental Results and Analysis

### 5.1. Drainages

Figure 5 compiles the distribution of the nonwetting fluid (green-colored) at the conclusion of specific drainage steps (1st, 5th, and 10th) in the tested micromodels. These micromodels represent different pore-space heterogeneities (COV values of 0, 0.25, and 0.5) and are subjected to two distinct flow rates (*Q* = 0.01 and 0.1 mL/min). The invasion of the nonwetting fluid at the end of the first drainage occurred primarily in the direction of the pressure gradient (i.e., parallel) when COV = 0. This trend was more prominent at *Q* = 0.1 mL/min (i.e., higher flow rate). On the other hand, the invasion of the nonwetting fluid occurred with more branches as the heterogeneity level increased (COV = 0 → 0.5), which implies that capillarity had a greater influence. Similar to the study results with Type I in Zhang et al. [3], the nonwetting fluid occupied the pore space more uniformly as the repetitive injection cycle continued at *Q* = 0.01 mL/min (i.e., lower flow rate). Interestingly, such a trend with the cyclic flow became less obvious with *Q* = 0.1 mL/min (higher flow rate).

In contrast to the observations for Type I, the invasion pattern became more similar between steps in the heterogeneous pore network at *Q* = 0.01 mL/min. That is, *R*_c_ was larger for the higher COV (e.g., *R*_c_ = 0.3–0.5 at COV = 0 vs. *R*_c_ = 0.5–0.7 at COV = 0.5; Figure 5c). On the other hand, this effect of the COV no longer manifested as the flow rate increased (Figure 5d). Given that *R*_c_ was larger than 0.2–0.3 for all tested models, it can be presumed that 0.2–0.3 may be the lower limit of *R*_c_ for the tested geometry (square solids) with the structural heterogeneity of COV = 0–0.5 in this 2-D environment.

### 5.2. Imbibitions (Forced)

Similarly to Section 5.1, the distributions of the wetting (colorless oil) and residual nonwetting fluid (green-colored) at the end of selected imbibition steps are compiled in Figure 6. Compared with the results from Type I in Zhang et al. [3], the distribution of the residual nonwetting fluid in Type II micromodels was more sporadic and random when COV = 0 (homogeneous). More of the nonwetting fluid also appeared to be stranded in the main domain of the pore space as the COV increased at both flow rates. However, like the Type I study, increasing the flow rate helped to reduce the residual nonwetting fluid in Type II (compare Figure 6a with Figure 6b). One important implication is that Type II appears to be more efficient than Type I in terms of the withdrawal of the nonwetting fluid (that is, a proxy for the withdrawal of air/hydrogen for an actual PM-CAES/UHS operation) under the given flow rates.

Similar to the drainage process, *R*_c_ was larger in the heterogeneous models at *Q* = 0.01 mL/min (*R*_c_ = 0.1–0.3 at COV = 0 vs. *R*_c_ = 0.35–0.65 at COV = 0.5; Figure 6c). Noticeably, even when the flow rate increased, *R*_c_ was still higher in the heterogeneous models (*R*_c_ = 0.1–0.2 at COV = 0 vs. *R*_c_ = 0.1–0.4 at COV = 0.5 when *Q* = 0.1 mL/min; Figure 6d). This trend is opposite to that in the Type I study. In addition, it is worth noting that *R*_c_ at *Q* = 0.1 mL/min was smaller than that at *Q* = 0.01 mL/min for the same pore structures. This supports our observation that the effect of the flow rate outweighs that of structural heterogeneity as the flow rate increases (compare Figure 6c with Figure 6d).

### 5.3. Sweep Efficiency

We utilized sweep efficiency and effective sweep efficiency as quantitative measures to evaluate the invasion of the nonwetting fluid [3]. Sweep efficiency (*E*_nw_) is determined by comparing the volume of the nonwetting fluid (*V*_nw_) to the total pore volume (*V*_total_) of the micromodel following drainage. Mathematically, *E*_nw_ is calculated as *E*_nw_ = *V*_nw_/*V*_total_. Conversely, effective sweep efficiency (*eE*_nw_) is determined by comparing the difference between the volume of the nonwetting fluid after drainage (*V*_nw_) and before drainage (*V*_rnw_) to the volume of the wetting fluid (*V*_w_) before drainage. Mathematically, *eE*_nw_ is calculated as *eE*_nw_ = (*V*_nw_ − *V*_rnw_)/*V*_w_.

The analysis results are summarized in Figure 7. It appears that the COV exerted a meaningful influence on the occupation of nonwetting fluid at a relatively lower flow rate. That is, both the sweep and effective sweep efficiencies increased as the COV intensified (0 → 0.25 → 0.5) at *Q* = 0.01 mL/min (Figure 7a,c). This implies that it was relatively easier for the nonwetting fluid to displace the wetting fluid in the heterogeneous micromodels when the capillarity effect was dominant. In contrast, the effect of the COV was no longer manifest at *Q* = 0.1 mL/min (i.e., higher flow rate), being consistent with the above observations for drainage (Section 5.1).

Indeed, both the sweep and effective sweep efficiencies were similar for all micromodels with different COV values at *Q* = 0.1 mL/min (Figure 7b,d). Rather, the main influential factor was the flow rate, which acted to elevate the overall occupation of the nonwetting fluid (compare *Q* = 0.01 vs. 0.1 mL/min). For example, the effective sweep efficiency was increased from *eE*_nw_ = 0.15–0.3 when *Q* = 0.01 mL/min to *eE*_nw_ = 0.35–0.45 when *Q* = 0.1 mL/min at COV = 0 (Figure 7c with Figure 7d). A noticeable difference from the Type I study results is the trivial impact of the COV at the higher flow rate owing to the different pore shapes and, thus, different aspect ratios (the average ratio of the width of the pore space to the width of the pore throat: Type I: 5–6 > Type II: ~1).

### 5.4. Residual Saturation

In a similar manner to sweep efficiency, we investigated both the absolute and effective residual saturation of the nonwetting fluid. The residual saturation (*S*_rnw_) is determined by comparing the volume of the nonwetting fluid at the end of each imbibition (*V*_rnw_) to the total pore volume (*V*_total_) of the micromodel after imbibition. Mathematically, *S*_rnw_ is calculated as *S*_rnw_ = *V*_rnw_/*V*_total_. Additionally, we defined the effective residual saturation (*eS*_rnw_) by comparing the volume of the residual nonwetting fluid (*V*_rnw_) to the volume of the nonwetting fluid just before imbibition (*V*_nw_). Mathematically, *eS*_rnw_ is calculated as *eS*_rnw_ = *V*_rnw_/*V*_nw_.

Noticeably, the residual saturation (both absolute and effective) was lowest in the homogeneous model (COV = 0) when *Q* = 0.01 mL/min (Figure 8a,c), which is exactly opposite to the Type I study results. This implies that the wider distribution of capillarity (i.e., COV → 0.5) might hinder the imbibition of the wetting fluid when the flow channels are rather straight with a low aspect ratio. The governance of the structural heterogeneity and the aspect ratio was persistent even with the increase in flow rate. That is, both the absolute and effective residual saturation of the nonwetting fluid was still lowest in the homogeneous model, which is then the same trend as Type I (Figure 8b,d).

Thus, overall, we can infer that the withdrawal efficiency of the nonwetting fluid may be better in the homogeneous pore structure when it belongs to the Type II type. Furthermore, the higher flow rate can yield an improved withdrawal efficiency for the same pore structure (compare Figure 8a with Figure 8b, and compare Figure 8c with Figure 8d), as long as the coning problem of the wetting fluid does not develop.

### 5.5. Flow Morphology

It can be interpreted that a larger normalized boundary length (*m*_0_) could suggest there are larger numbers of smaller fluid ganglia in the flow domain, as noted by Zhang et al. [3]. In this study, it was clearly observed that *m*_0_ decreased as the structural heterogeneity of the pore-network model was intensified (COV = 0 → 0.25 → 0.5) during both drainage and imbibition under the tested flow rates (Figure 9). This is the same trend as that in the Type I study by Zhang et al. [3].

We infer that a more randomized distribution of the square solids in the main flow domain, associated with the low aspect ratio of the flow channels (≈1), enabled longer and more connected ganglia of the nonwetting fluid, which yielded a smaller value of *m*_0_. In addition, like in the Type I study, *m*_0_ was quite stable in each micromodel during the repetitive drainage–imbibition cycles. The influence of flow rate (*Q*) on *m*_0_ also seemed insignificant for any given COV, although the effective sweep efficiency improved substantially with a higher *Q* (Figure 7).

### 5.6. Fluid Pressure

Lastly, we monitored the largest pressure drop across the model (Δ*P*_max_) during the repetitive drainage–imbibition tests (Figure 10). A larger pressure gradient was needed for more heterogeneous models (COV → 0.5) at *Q* = 0.01 mL/min (e.g., lower flow rate) during both the drainage and imbibition processes. This is the opposite result from that of the Type I study by Zhang et al. [3]. Given that the average of the aspect ratios in these Type II models was almost 1 (cf., Type I: 5–6), the distribution of capillarity in each flow channel, associated with the viscosities of the fluids, might be the main contributor to the pressure drop during the fluid flow, leading to the opposite results compared with the Type I study where the Haines jump was pervasive. However, this trend with the COV again vanished as *Q* increased to 0.1 mL/min (e.g., higher flow rate).

Overall, Δ*P*_max_ was higher during imbibition than drainage, particularly at high *Q* values (compare Figure 9c with Figure 9d), which is the same trend from the Type I study. It implies that more work (=pressure × volume) would be required for the wetting fluid to displace the nonwetting fluid for a given volume of fluid flow when imbibition is forced rather than spontaneous. This also became more prominent with an increase in the flow rate (*Q*).

We infer that the randomly distributed residual (i.e., trapped) nonwetting fluid—and thus a more tortuous travel path for the highly viscous wetting fluid during imbibition—contributed to this observation. Therefore, we may predict that a higher pressure gradient would be required during the withdrawal of compressed air/hydrogen back to the surface unit in operation (like the PM-CAES/UHS), and attention should be paid to this phenomenon to reduce the possibility of the coning problem.

Lastly, just like the findings from the Type I study, the maximum pressure drop (Δ*P*_max_) exhibited a 2-4 times increase when the flow rate (*Q*) was multiplied by a factor of 10. This can be observed by comparing Figure 10a to Figure 10c, as well as Figure 10b to Figure 10d. Once again, this demonstrates that the increase in viscous pressure loss is not directly proportional to the applied flow rate when both phases are flowing simultaneously. Table 3 summarizes all the experimental results for *R_c_*, *E*_nw_, *eE*_nw_, *S*_rnw_, *eS*_rnw_, *m*_0_, and Δ*P*_max_.

## 6. Discussion

### 6.1. Impact of COV (Pore-Space Heterogeneity), Q (Flow Rate), and Aspect Ratio (Pore Shape)

The feature of a nonwetting fluid preferring to invade a larger pore throat first is more prominent at a low flow rate. It can explain the higher sweep efficiency with larger COVs in Type II models at *Q* = 0.01 mL/min in this study (Figure 7). However, similar to the observations from the Type I study, a higher flow rate (e.g., *Q* = 0.1 mL/min in this study) dwarfs the influence of pore-space heterogeneity; consequently, in association with the small aspect ratio (≈1), this acts as a main factor that decides the sweep efficiency of a nonwetting fluid during drainage. In detail, the higher flow rate, and thus larger capillary pressure, helps to push the nonwetting fluid through more flow channels with smaller widths. Consequently, both injection and withdrawal efficiencies were improved. This was observed for both Type I and II geometries. In addition to this, the square pore shape, and thus small aspect ratios and straight flow paths, results in the most efficient withdrawal of the nonwetting fluid in the homogeneous pore structure for all the tested flow rates and COVs (Figure 8).

Overall, the COV and pore shape may exert a significant influence at a low flow rate, but a high flow rate can effectively outweigh those two factors and, thus, primarily govern the sweep efficiency and residual saturation of the nonwetting fluid. In addition, the flow morphology, sweep efficiency, residual saturation, and required pressure gradient may not severely fluctuate during the repeated drainage–imbibition processes, becoming stabilized after 4–5 cycles, regardless of the COV, flow rate, and pore shape. Lastly, a larger pressure gradient may be needed as the COV increases in the Type II structure with the small aspect ratio, which is the opposite in the Type I structure.

The impact of the surface chemistry of the porous material on the sweep efficiency and residual saturation during the repetitive drainage–imbibition processes was not considered in this study. A follow-up study that can integrate such an impact of the surface chemistry with the above factors can improve further our understanding on this topic.

### 6.2. Implications for Air/Gas Storage

The implications for CAES in porous media have been discussed in depth by Zhang et al. [3]. However, owing to the different pore shapes, it is worth discussing the implications for either PM-CAES or UHS using geologic storage reservoirs of the Type II geometry here. The same implications regarding gas injection, flow rate, and geometrical heterogeneity can be referred to in Zhang et al. [3].

*Sweep Efficiency*. The average *E*_nw_ during the repetitive drainage–imbibition cycles in Type II with different COVs was around 0.45 at *Q* = 0.1 mL/min (Figure 7b). The permeability of the micromodel used in this study (4.0–6.4 Darcy) is higher than the permeability range (0.5–1 Darcy) found in the literature [24,25,26,27]. Therefore, once again, it is implied that the real sweep efficiency of PM-CAES/UHS may be lower than the anticipated ranges mentioned in existing literature.

*Air Withdrawal*. Overall, we can infer that the withdrawal efficiency of air/gas may be better in the homogeneous pore structure when it belongs to the Type II geometry. Compared with the Type I geometry, the relatively smaller flow resistance in the main pore-network domain during imbibition may imply a higher possibility of the water-coning problem into the wellbore for a short withdrawal time. In this regard, the issue of water-coning needs more attention in the further investigation of PM-CAES/UHS in carbonate reservoirs as energy storage media. Lastly, one of the key performance indices, the efficiency of air injection and withdrawal, would be stabilized after 4–5 cycles during the actual implementation of geologic subsurface energy storage.

## 7. Conclusions

As a companion to the study by Zhang et al. [3], we conducted a follow-up study with microfluidic pore-network devices with a square solid shape (Type II) to further advance our understanding of the effect of the pore shape (aspect ratio), pore-space heterogeneity (COV), and flow rates (*Q*), on the repetitive two-phase fluid flow in general porous media. The influence of pore shape and pore-space heterogeneity on the examined outcomes, including the drainage and imbibition patterns, the similarity of those patterns between repeated steps, the sweep efficiency and residual saturation of nonwetting fluid, and fluid pressure, were observed to be more prominent when the flow rate was low (e.g., *Q* = 0.01 mL/min in this study). On the other hand, a higher flow rate (e.g., *Q* = 0.1 mL/min in this study) appeared to outweigh those factors for the Type II structure, owing to the low aspect ratio (~1). It was also suggested that the flow morphology, sweep efficiency, residual saturation, and required pressure gradient may not severely fluctuate during the repeated drainage–imbibition processes, becoming stabilized after 4–5 cycles, regardless of the aspect ratio, COV, or *Q*.

Specifically, similar to the study results with Type I, the nonwetting fluid occupied the pore space more uniformly as the repetitive injection cycle continued at *Q* = 0.01 mL/min (i.e., lower flow rate). Interestingly, such a trend with the cyclic flow became less obvious with *Q* = 0.1 mL/min (higher flow rate). The distribution of the residual nonwetting fluid in Type II micromodels was more sporadic and random when COV = 0 (homogeneous). In addition, more of the nonwetting fluid appeared to be stranded in the main domain of the pore space as the COV increased at all tested flow rates.

Overall, we can infer that the withdrawal efficiency of the nonwetting fluid may be better in the homogeneous pore structure when it belongs to the Type II geometry. However, a higher flow rate can improve both the injection and withdrawal efficiencies at a given pore structure of Type II. Lastly, a larger pressure gradient may be needed as the COV increases in Type II, which is the opposite of the Type I result. Compared with Type I, the relatively smaller flow resistance in the main pore-network domain during imbibition may imply a higher possibility of the water-coning problem into the wellbore for a short withdrawal time.

## Figures and Tables

**Figure 1 micromachines-14-01441-f001:**
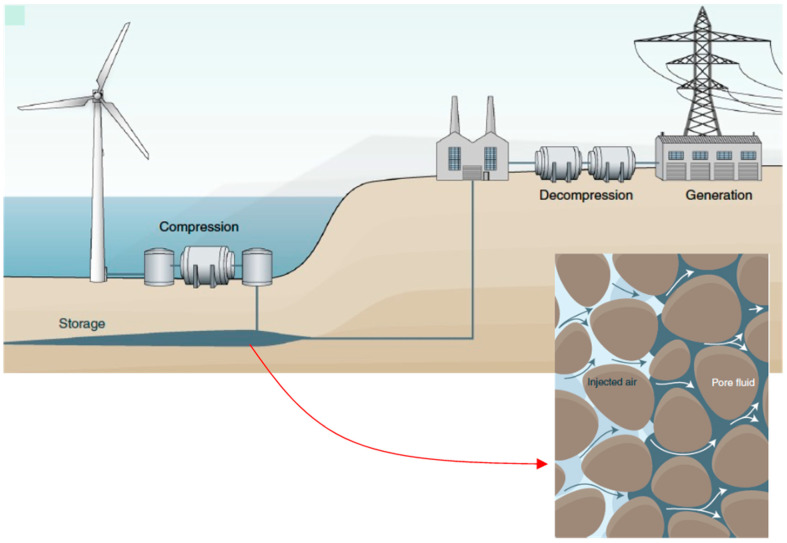
Conceptual diagram of compressed-air energy storage in porous media (PM-CAES) using a porous geologic formation as a storage medium. The image is edited from Bentham [4].

**Figure 2 micromachines-14-01441-f002:**
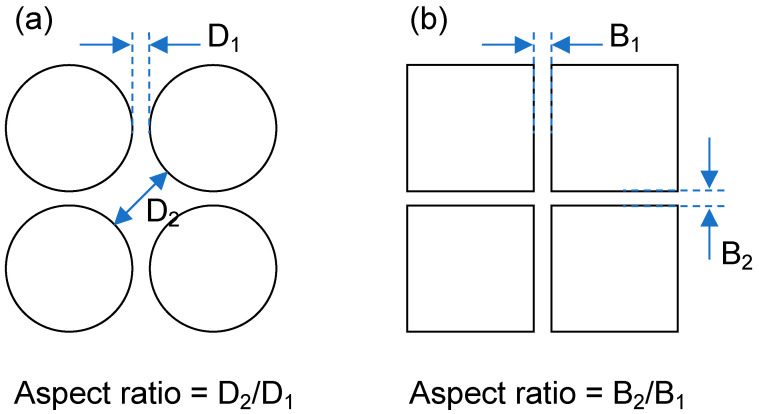
Definition of the aspect ratio for (**a**) Type I and (**b**) Type II geometry in this study.

**Figure 3 micromachines-14-01441-f003:**
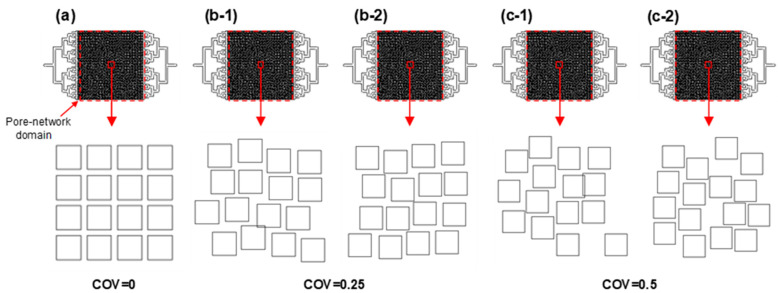
The microfluidic pore-network model’s geometry, modified by varying the coefficients of variation (COV) for the width of the pore-throats. The overall size of the pore-network domain was 20 mm × 20 mm, and the width of the square solid was 320 μm. In the case of the homogeneous pore-network model (designated as “**a**” with COV = 0), the pore-throat width was consistently 80 μm. For the heterogeneous pore-network models (designated as “**b-1**,” “**b-2**” with COV = 0.25, and “**c-1**,” “**c-2**” with COV = 0.5), the minimum pore-throat width was set to 20 μm. It is important to note that two distinct microfluidic pore-network models were created to represent COV values of 0.25 and 0.5, respectively. In cases where the square solids overlapped, they were combined into a single entity. The effective porosity was approximately 0.36 for models with COV values of 0 and 0.25, while for models with a COV of 0.5, the effective porosity was around 0.38.

**Figure 4 micromachines-14-01441-f004:**
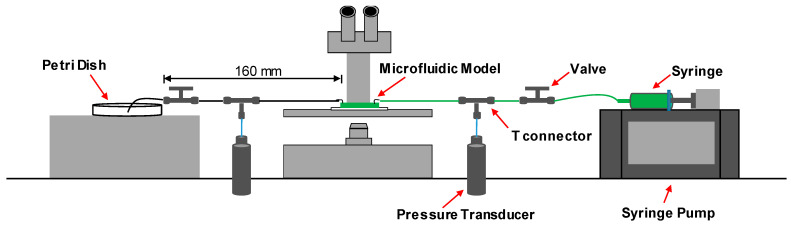
The experimental arrangement involved the implementation of repetitive drainage-imbibition cycles in the microfluidic device. It should be noted that water, which was colored green, served as the nonwetting fluid, while mineral oil, which had no color, acted as the wetting fluid.

**Figure 5 micromachines-14-01441-f005:**
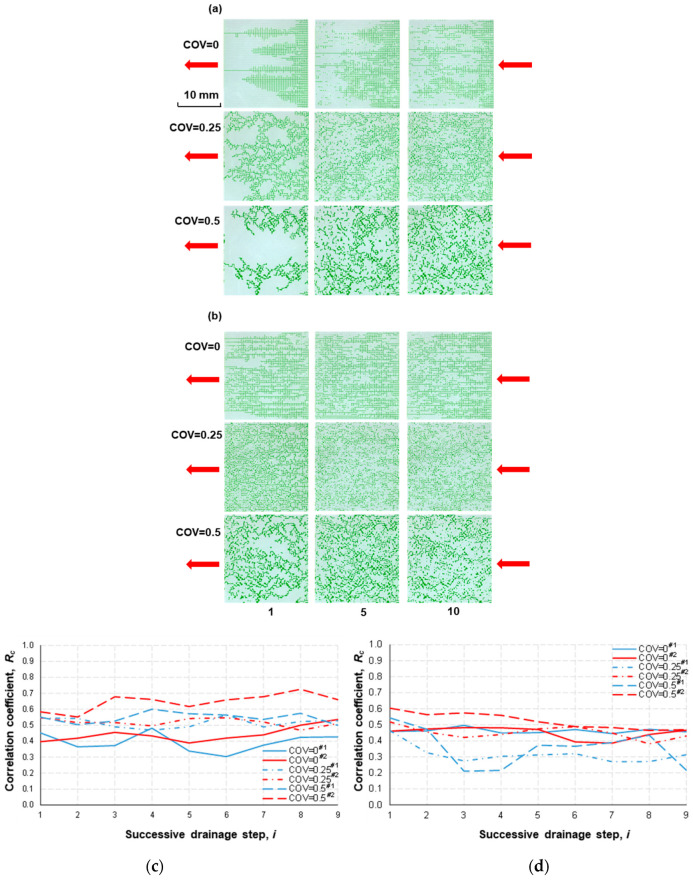
Distribution of the nonwetting fluid (green-colored) at the end of 1st, 5th, and 10th drainage steps: (**a**) volumetric flow rate *Q* = 0.01 mL/min and (**b**) 0.1 mL/min in different microfluidic pore-network models (the number below each column of images indicates the turn of cycles). The correlation coefficient (*R*_c_) was calculated to assess the similarity between successive images of the nonwetting fluid distribution at the end of each drainage step. This analysis was performed separately for the two flow rates: (**c**) *Q* = 0.01 mL/min and (**d**) 0.1 mL/min. The x-axis label “*i*” represents the correlation coefficient, *R*_c(i)_, between the *i*th and (*i* + 1)th drainage steps. The red arrows in the images indicate the direction of injection for the nonwetting fluid. It should be noted that imbibition of the wetting fluid (colorless) occurred between drainage cycles. The designations COV = 0^#1^ and 0^#2^ represent two independent tests conducted using the homogeneous model. Similarly, COV = 0.25^#1^ and 0.25^#2^ correspond to two separate tests conducted using (b-1) and (b-2) models shown in Figure 3. Likewise, COV = 0.5^#1^ and 0.5^#2^ refer to the tests conducted using (c-1) and (c-2) models. The legend includes identifiers #1 and #2, which represent different testing numbers.

**Figure 6 micromachines-14-01441-f006:**
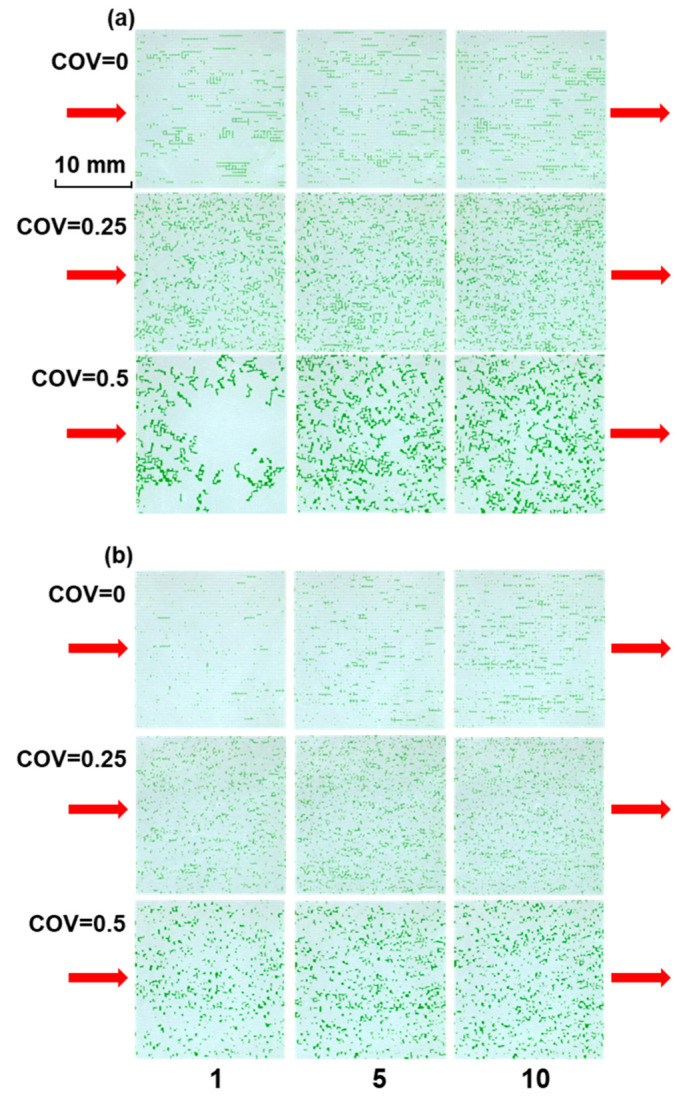
Distribution of the wetting fluid (colorless) and residual nonwetting fluid (green-colored) at the end of 1st, 5th, and 10th imbibition steps: (**a**) volumetric flow rate *Q* = 0.01 mL/min and (**b**) 0.1 mL/min in different microfluidic pore-network models (the number below each column of images indicates the turn of cycles). The correlation coefficient (*R*_c_) was calculated to assess the similarity between successive images of the wetting fluid distribution at the end of each imbibition step. This analysis was performed separately for the two flow rates: (**c**) *Q* = 0.01 mL/min and (**d**) 0.1 mL/min. The x-axis label “*i*” represents the correlation coefficient, *R*_c(i)_, between the *i*th and (*i* + 1)th imbibition steps. The red arrows in the images indicate the direction of flow for the wetting fluid. It should be noted that drainage of the nonwetting fluid (green-colored) occurred between imbibition cycles. The designations COV = 0^#1^ and 0^#2^ represent two independent tests conducted using the homogeneous model. Similarly, COV = 0.25^#1^ and 0.25^#2^ correspond to two separate tests conducted using (b-1) and (b-2) models shown in Figure 3. Likewise, COV = 0.5^#1^ and 0.5^#2^ refer to the tests conducted using (c-1) and (c-2) models. The legend includes identifiers #1 and #2, which represent different testing numbers.

**Figure 7 micromachines-14-01441-f007:**
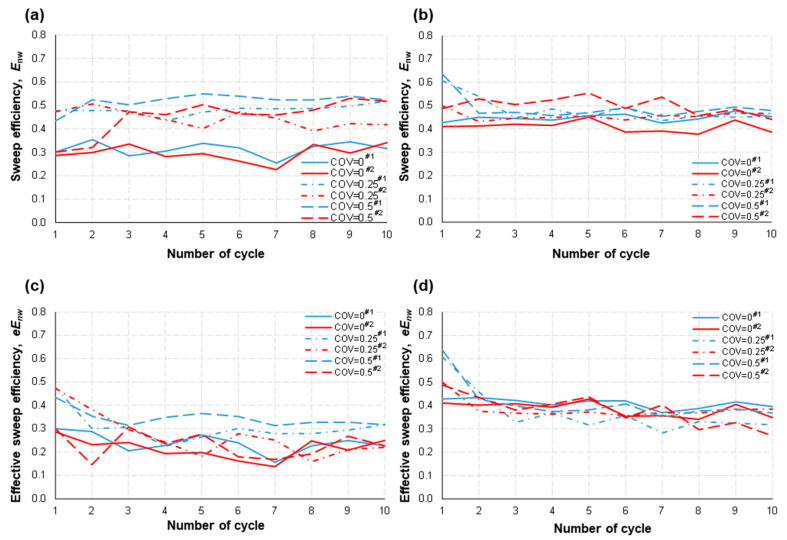
The sweep efficiency of the nonwetting fluid, *E*_nw_, was determined for two different volumetric flow rates: (**a**) *Q* = 0.01 mL/min and (**b**) 0.1 mL/min. Additionally, the effective sweep efficiency of the nonwetting fluid, *eE*_nw_, was evaluated for (**c**) *Q* = 0.01 mL/min and (**d**) 0.1 mL/min at the conclusion of each drainage step. The legend includes identifiers #1 and #2, which represent different testing numbers.

**Figure 8 micromachines-14-01441-f008:**
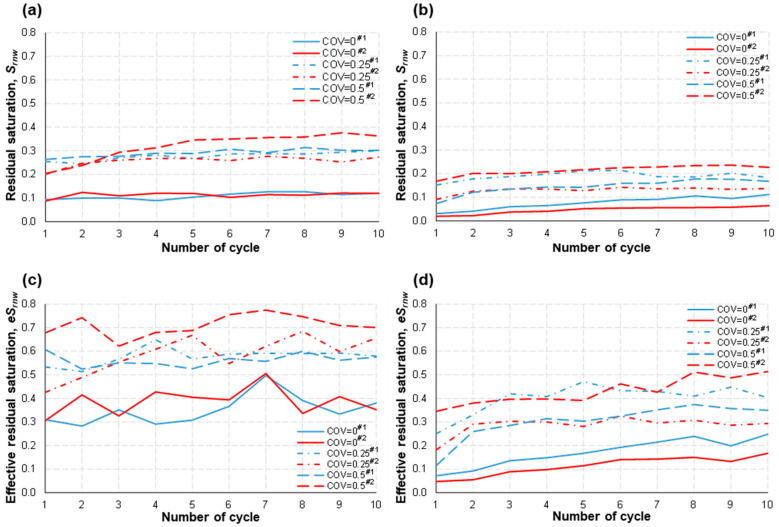
The residual saturation of the nonwetting fluid, *S*_rnw_, was determined for two different volumetric flow rates: (**a**) *Q* = 0.01 mL/min and (**b**) 0.1 mL/min. Additionally, the effective residual saturation of the nonwetting fluid, *eS*_rnw_, was calculated for (**c**) *Q* = 0.01 mL/min and (**d**) 0.1 mL/min at the conclusion of each imbibition step. The legend includes identifiers #1 and #2, which represent different testing numbers.

**Figure 9 micromachines-14-01441-f009:**
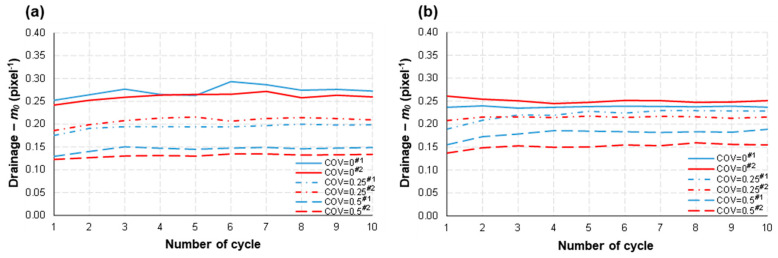
The normalized Minkowski functionals, *m*_0_ (which represents the ratio of boundary length to the covered area of the nonwetting fluid), were computed for two different volumetric flow rates: (**a**) *Q* = 0.01 mL/min and (**b**) 0.1 mL/min, at the conclusion of each drainage step. Additionally, the normalized m_0_ function was calculated for (**c**) *Q* = 0.01 mL/min and (**d**) 0.1 mL/min, at the end of each imbibition step. The legend includes identifiers #1 and #2, which represent different testing numbers.

**Figure 10 micromachines-14-01441-f010:**
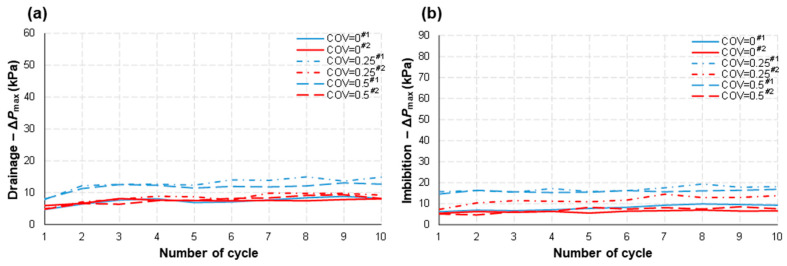
The maximum pressure difference, Δ*P*_max_, applied to the pore-network domain to sustain a volumetric flow rate of *Q* = 0.01 mL/min was measured during (**a**) drainage and (**b**) imbibition. Similarly, Δ*P*_max_ was determined for *Q* = 0.1 mL/min during (**c**) drainage and (**d**) imbibition. These measurements were conducted for various structural heterogeneities represented by different coefficients of variation (COV = 0, 0.25, and 0.5). The legend includes identifiers #1 and #2, which represent different testing numbers.

**Table 1 micromachines-14-01441-t001:** Geometric features of the pore-network micromodels used in this study.

Parameters	Figure 3b-1	Figure 3b-2	Figure 3c-1	Figure 3c-2
COV = *σ*_w_/*w*	0.25	0.25	0.5	0.5
Mean width, *w* (μm)	80	80	80	80
Standard deviation, *σ*_w_ (μm)	20	20	40	40
Minimum width, *w*_min_ (μm)	20	20	20	20
Height, *h* (μm)	100	100	100	100
Effective porosity	0.36	0.36	0.38	0.38
Permeability (Darcy)	4.2	5.9	4.0	6.4

**Table 2 micromachines-14-01441-t002:** Properties of fluids used in this study.

Mineral Oil *		Water		Water–Oil
Viscosity (Pa·s)	Density (kg/m^3^)	Viscosity (Pa·s)	Density (kg/m^3^)	Interfacial Tension (mN/m)
4.2 × 10^−2^	830	1.0 × 10^−3^	1000	52

* Viscosity and density values of the mineral oil are at 25 °C.

**Table 3 micromachines-14-01441-t003:** Summary of the experimental observations on the repetitive two-phase fluid flow in microfluidics pore-network models with different pore-space heterogeneities (COV) and flow rates (*Q*).

*Q*	0.01 mL/min			0.1 mL/min		
COV	0	0.25	0.5	0	0.25	0.5
*R_c_*—Drainage	0.3–0.55	0.45–0.55	0.5–0.7	0.4–0.5	0.25–0.5	0.2–0.6
*R_c_*—Imbibition	0.1–0.35	0.25–0.45	0.35–0.65	0.1–0.2	0.1–0.4	0.1–0.4
*E* _nw_	0.25–0.35	0.4–0.5	0.3–0.55	0.35–0.45	0.45–0.6	0.45–0.55
*eE*_nw_ *	0.15–0.3	0.15–0.4	0.15–0.35	0.35–0.45	0.3–0.45	0.3–0.45
*S* _rnw_	0.1–0.15	0.25–0.3	0.2–0.35	0.05–0.1	0.1–0.2	0.1–0.25
*eS*_rnw_ *	0.3–0.5	0.5–0.7	0.55–0.8	0.05–0.25	0.3–0.45	0.25–0.5
*m*_0_—Drainage	0.25–0.3	0.17–0.25	0.13–0.15	0.24–0.25	0.22–0.23	0.15–0.18
*m*_0_—Imbibition	0.28–0.32	0.22–0.25	0.14–0.17	0.3–0.35	0.26–0.27	0.2–0.23
Δ*P*_max_—Drainage	5–8 kPa	5–15 kPa	5–14 kPa	30–35 kPa	22–35 kPa	10–33 kPa
Δ*P*_max_—Imbibition	6–10 kPa	8–20 kPa	5–15 kPa	40–47 kPa	25–50 kPa	35–80 kPa

* After the first drainage–imbibition cycle.

## Data Availability

The data presented in this study are available on request from the corresponding author.

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
