# Peer review of "Experimental Study: The Effect of Pore Shape, Geometrical Heterogeneity, and Flow Rate on the Repetitive Two-Phase Fluid Transport in Microfluidic Porous Media"

_micromachines, 2023, doi:10.3390/mi14071441_

Round 1
Reviewer 1 Report
This paper presents a comprehensive experimental study that investigates the impact of pore shape, geometry, and flow rate on repetitive two-phase fluid transport in microfluidic devices. It is important to note that this work heavily relies on a previous research study by Zhang et al. (2021) and therefore should strive to demonstrate independent and innovative ideas before it can be considered for publication. Also, the experiment design and narrative of the paper could benefit from improvements, the study's thoroughness contributes significantly to the field. To further enhance the paper's value, I recommend incorporating more innovative ideas into the research. Addressing these points through major revisions will possibly improve the paper's potential for successful publication.
Major:
1. As a follow-up study, this paper falls short in terms of presenting innovative ideas that can contribute significantly to the field. Furthermore, the absence of incorporating heterogeneity patterns raises concerns regarding the justification for representing real porous media in the context of PM-CAES and UHS. Previous research has already explored ways to better represent natural porous media using heterogeneous patterns, and the method demonstrated in this study appears outdated.
a) Improved sweep efficiency due to foam flooding in a heterogeneous microfluidic device’(https://doi.org/10.1016/j.petrol.2018.01.042)
b) Visualization study of CO2-EOR in carbonate reservoirs using 2.5D heterogeneous micromodels for CCUS (https://doi.org/10.1016/j.fuel.2022.125533)
2. The paper primarily focuses on the effect of shape, pore size, and geometry, which may not directly translate to real-world applications. It is important to consider that the surface chemistry of the porous material can significantly affect factors such as sweeping efficiency and residual saturation. Therefore, further research or details should be conducted to explore the impact of surface chemistry.
3. This paper lacks the necessary references to help readers follow the previous work effectively, which can make it challenging for them to understand the context and build upon prior knowledge. It is crucial to provide proper credit and acknowledgment to the relevant prior works to establish a strong foundation for the current study. Below are a few examples of mis-cited references, highlighting just a few presents in the paper.
a) Page 2 lines 48-49 pioneer work on the corresponding energy storage technology (batteries, pumped hydro storage, compressed air energy storage, hydrogen storage, flywheels, and thermal storage) should be included for reference. Although it does not necessary to mention.
b) Page 2 lines 53-53 PM-CAES and UHS are the potential application value of this research work, the pioneer work for those core concepts should be cited.
c) Page 3 lines 82-83 when the authors refer to the capillary number (Ca) and viscosity ratio (M) reference should credit to the original equation instead of the research work they followed.
d) Page 3 lines 87-88 the author stated, ‘It is widely accepted that the pattern of fluid flow in porous media will be comparable if Ca and M are in the same order for any given operation’ Need a citation to justify the statement
e) Page 4 lines 132-133 the author claimed that ‘Type I was designed to mimic regularly shaped intergranular pores of general sandstones with a randomized distribution of circular solids in the pore network.’ Corresponding citations should be inserted to justify the statements.
4. it is crucial for the authors to provide more detailed experiment details. The inclusion of comprehensive experimental procedures, methodologies, and protocols will allow other researchers to reproduce the study and validate the findings.
a) In Figure 3, specifically in panels B-1 and C-1, the design of the porous media exhibits an overlap between the squared patterns. It is important to clarify whether these designs are CAD representations or actual pattern representations. The presence of overlapping squares could potentially be misleading, as it may not accurately reflect the intended design or the real-world scenario. It is essential to address this issue and provide clarification to ensure the accurate interpretation of the results and prevent any potential misinterpretations.
b) The author claims to have produced a hydrophobic PDMS-based microfluidic channel. However, the fabrication method for achieving hydrophobicity is not adequately described and simply referencing a source is insufficient. It is essential to provide detailed information on the fabrication process employed to make the microfluidic channel hydrophobic, including any surface modification techniques or coatings used.
c) Furthermore, the characterization of the materials and porous media is limited. To provide a comprehensive understanding of the system, additional characterization data should be included. Specifically, for wettability characterization, contact angle measurements should be provided to quantify the hydrophobicity of the surfaces.
d) Additionally, it would be beneficial to include a magnified picture of the microfluidic device to allow for a more detailed examination of the pattern and structural features. This would provide better visualization and aid in understanding the design and arrangement of the microfluidic system.
e) The height of the microfluidic channel is stated to be 100 microns, and the pore size of the porous media is relatively small compared to other research in the field. This difference in height and pore size could lead to the heterogeneous distribution of height and depth within the system. To address this issue, a profilometry analysis should be performed to assess and present the surface topography and ensure the accuracy and consistency of the dimensions throughout the microfluidic channel and porous media.
5. The experiment quality in this research has the potential to be improved. To enhance the overall experiment quality, the following suggestions are provided:
a) Regarding image processing line 100 the value of a pixel is 0 or 1, having a greyscale index that is continuous could help better quantify the residual fluid.
b) PDMS piece is bounded with a glass slide to fabricate the microfluidic device, the wettability of pores media should be considered.
c) The deformation of PDMS in contact with oil should be considered since it could alter the pore size and permeability of the channel (eg: https://link.springer.com/article/10.1007/s42452-020-03288-8)
Minor:
1. Line 122 what is the build-in function in MATLAB, is it a specific module or the author’s own code base?
2. In the table-1 how is permeability measured? Details should be included in the experiment section.
3. In line 230, the author claim that water was a nonwetting fluid (green-colored). What kind of dye the authors used, and what is the concentration? Whether it is interfacial active or not? That information should be included.
The authors of this research work demonstrate a high level of professional capability and logic fluency in presenting their findings. They effectively utilize clear and concise language to communicate their research methodology, results, and conclusions. The logical flow of their presentation allows for a coherent understanding of the research process and outcomes.
Author Response
Please see our responses in the attached word file.

Reviewer 2 Report
The study conducted microfluidic experiments using square-shaped (Type II) pore-network devices to investigate the impact of pore shape, pore-space heterogeneity, and flow rates on repetitive two-phase fluid flow in porous media. They found that the influence of pore shape and heterogeneity was more significant at low flow rates, while higher flow rates minimized these effects. The study also suggests that certain flow characteristics stabilize after 4-5 cycles, regardless of pore shape, heterogeneity, and flow rates, and discusses the implications of the findings for geologic subsurface energy storage applications.
The article is well written and presented, I consider it should be accepted for publication.
Some minor questions are:
Could you provide more details on the methodology employed to seal the PDMS device in the experimental setup, considering the utilization of positive pressure? Additionally, could you elaborate on the specific instances where leaks were observed in the PDMS device during the experimentation?
The study suggests that flow morphology, sweep efficiency, residual saturation, and required pressure gradient stabilize after 4-5 cycles of drainage-imbibition, regardless of aspect ratio, pore-space heterogeneity, and flow rates. Could you discuss the implications of these findings for practical applications, such as geologic subsurface energy storage?
It is mentioned that the distribution of residual nonwetting fluid in Type II micromodels becomes more sporadic and random as pore-space heterogeneity (COV) increases. Could you provide further explanations on how the COV affects the distribution and withdrawal efficiency of the nonwetting fluid in the Type II geometry?
The discussion notes that a higher flow rate improves both injection and withdrawal efficiencies for the Type II structure. Could you provide insights into the mechanisms behind this improvement and how it relates to the specific characteristics of the Type II geometry?
Author Response

(The authors gave the same response as above.)

Round 2
Reviewer 1 Report
After carefully reviewing the revised manuscript, I must commend the authors on their diligent efforts in addressing the previously mentioned concerns. The changes they have made have enhanced the overall quality of the paper. Considering these improvements, I would like to suggest the publication of this manuscript in Micromachines.
The quality of English has been improved would suggest considering for publication in Micromachines.